# Efficacy and effectiveness of SARS-CoV-2 vaccines for death prevention: A protocol for a systematic review and meta-analysis

Anete Trajman[1,2☯]*, Sophie Lachapelle-Chisholm[1☯]*, Théodora Zikos[3☯], Guilherme Loureiro Werneck[4,5‡], Andrea Benedetti[1‡]

1 Research Institute, McGill University Health Center, Montréal, QC, Canada, 2 Faculdade de Medicina, Federal University of Rio de Janeiro, Rio de Janeiro, Brazil, 3 Institut National d'Excellence en Santé et en Services Sociaux, Montreal, QC, Canada, 4 Instituto de Medicina Social Hesio Cordeiro, State University of Rio de Janeiro, Rio de Janeiro, Brazil, 5 Instituto de Estudos de Saúde Coletiva, Federal University of Rio de Janeiro, Rio de Janeiro, Brazil

☯ These authors contributed equally to this work.
‡ LW and AB also contributed equally to this work.
* sophie.lachapelle-chisholm@rimuhc.ca (SLC); anete.trajman@medicina.ufrj (AT)

**Data Availability Statement:** Not applicable. This is a protocol, no results will be available.

## Abstract

### Background

There is consistent evidence that SARS-CoV-2 vaccines have statistical and clinical significant efficacy to prevent incident and severe cases of COVID-19, although different outcomes were analyzed and different risk reductions were observed. However, randomized control trials (RCT) were not designed or powered to assess whether the vaccines prevent deaths, even though this was a secondary or exploratory outcome across many studies. Early real-world observational data suggest that these vaccines are highly effective in reducing hospitalization and all-cause mortality. Our objective is to summarize and appraise—the existing evidence on the efficacy and real-world effectiveness of all SARS-CoV-2 vaccines currently approved for full or limited use to prevent all-cause and COVID-19-attributed mortality.

### Methods

The **population** consists of persons with a record of vaccination status and the outcome of interest. Randomized controlled trials, comparative cohort and case-control studies reporting vaccination with any of the vaccines approved (**intervention**) will be eligible. The primary **outcome** will be all cause deaths. COVID-19-attributed deaths and deaths attributable to the vaccination (adverse event deaths) will be secondary outcomes. We will **compare** deaths occurring in vaccinated persons versus those non-vaccinated or having received placebo. Studies in any language will be eligible. Two independent reviewers will screen for inclusion and assess quality of studies using the Cochrane Risk of Bias 2 and the ROBINS-1 tool, as appropriate. Hazard ratios will be calculated. Assessment of statistical heterogeneity amongst the studies will be done using $I^2$ and prediction intervals, as well as visual inspection of the forest plots. Publication bias will be assessed using a funnel plot and

**Funding:** The authors received no specific funding for this work.

**Competing interests:** The authors have declared that no competing interests exist.

Egger statistical test if we have more than 10 studies in a forest plot. We have followed the PRISMA-Protocol checklist for the current protocol, which is registered at Prospero (York University, CRD42021262211).

## Background

The emergence of human infection with SARS-CoV-2 was first observed in late 2019, when cases of unexplained pneumonia were reported in Wuhan, China [1]. The human disease caused by SARS-CoV-2 is known as COVID-19. The World Health Organization (WHO) declared a COVID-19 pandemic on March 11[th], 2020. Successive waves of high transmission and the emergence of new variants in several countries ensued [2]. As of January 6th, 2022, 5,462,631 deaths from COVID-19 had been reported worldwide, though this is likely an underestimation [3]. COVID-19 is thus the leading cause of death from an infectious disease worldwide.

In contrast with the multi-year long developmental process of previous vaccines, less than a year after the SARS-CoV-2 genome sequence was described on January 10th, 2020 [4], new vaccines have already been developed. Using different technologies, these vaccines have been tested for safety and efficacy in preventing SARS-CoV-2 infection and hospitalization, with the first vaccine having been implemented as early as December 20th, 2020 [5]. Currently, there are 27 vaccines undergoing Phase I safety trials, 30 undergoing combined Phase I and II trials, 9 undergoing expanded safety Phase II trials, 10 undergoing combined Phase II and III trials, 36 undergoing large scale efficacy Phase III trials, 9 undergoing Phase IV trials, with 23 vaccines approved for full use by several regulatory agencies [6]. In addition, 9 vaccines have been abandoned [7].

In a scoping review, we found that there is consistent evidence that SARS-CoV-2 vaccines —whichever the employed technology—have statistical and clinical significant efficacy to prevent cases of COVID-19, although different outcomes were analyzed and different risk reductions were observed [8–12]. However, these randomized control trials (RCT) were not designed or powered to assess whether the vaccines prevented deaths, even though this was a secondary or exploratory outcome across many studies. When isolated, none of these RCT had the power to ascertain whether the risk of death was reduced by the intervention.

Early observational data from different populations and settings in which the vaccines have been implemented have shown that vaccines are highly effective in reducing hospitalization and all-cause mortality [5, 11–17]. (5,–However, along with the intrinsic limitations with observational studies, some researchers have used an ecological design [18], which has led to quality of evidence that is heterogeneous. Important possible biases found with observational studies of vaccine effectiveness include confounding due to healthcare-seeking behaviour, exposure risk, misclassification of outcomes due to diagnostic errors, prior SARS-CoV-2 infection, and spurious inferences of waning [19]. Despite these limitations, the WHO recommends performing vaccine effectiveness observational studies to better understand the impact of large-scale vaccination on COVID-19 mortality, as RCTs have limited evidence [19]. These studies, according to the WHO, can have a high impact for public health decision-makers [19].

No systematic reviews on the efficacy and effectiveness of SARS-CoV-2 vaccines for preventing all-cause mortality were found in our scoping review. One published systematic review evaluated safety and surrogates of efficacy such as neutralizing antibodies, but mortality was not one of the endpoints [20]. We found seven registered ongoing systematic review protocols on the safety and efficacy of SARS-CoV-2 vaccines. One [21] will evaluate prevention of any

form of SARS-CoV-2 infection as the efficacy outcome and will only include evidence from single- or double-blinded control trials. The second will be a living mapping of research and network meta-analysis of multiple interventions to prevent the spreading of the disease [22]. Another living systematic review will analyze efficacy endpoints including death, but only observational studies will be included for adverse event analyses [23]. And another systematic review and network meta-analysis will attempt to compare the efficacy, effectiveness and safety of various COVID-19 vaccines using data from RCT studies and COVID-19 death will be one of the main outcomes [24]. Thus, pooled efficacy and real-world effectiveness of SARS-CoV-2 vaccination for all-cause mortality is not yet being evaluated.

Our objective is to summarize and appraise—and, if possible, pool—the existing evidence on the efficacy and real-world effectiveness of all SARS-CoV-2 vaccines currently approved for use for the prevention of all-cause and COVID-19-attributed mortality. We intend to do so by including RCTs, as well as cohort and case-control studies in our review. We hypothesize that the summarized data will suggest an association between vaccine use and a reduction of all-cause mortality.

## Objectives

The following PICO question will guide our methods: Compared to the non-vaccinated (or placebo-vaccinated) population, what is the efficacy and effectiveness of the currently available for use SARS-CoV-2 vaccines on all-cause mortality among individuals aged 12 years or over, in any setting worldwide, as of the first day of the first dose received?

- Disease: PCR-Rt or viral culture confirmed SARS-CoV-2 infection/COVID-19.

- Population: persons over 12 years old.

- Settings: all settings included (both high-(HIC) and low- and medium-income countries (LMIC), community, short and long-term care, remote, hospital settings).

- Interventions: any SARS-CoV-2 vaccination currently available for use in any country (26vaccines using varying technologies have been approved for full use in at least one country by several regulatory agencies as of 4 December 2021, see Appendix I in S1 Appendix).

- Comparators: non exposure to SARS-CoV-2 vaccination, placebo.

- Outcome: Primary outcome is all-cause mortality after 1st dose of vaccination, and 2nd dose of vaccination, if reported. The reasons for this choice are the imprecision of attribution of deaths to COVID-19, especially in LMIC [3]. Secondary outcomes are COVID-confirmed deaths, and death as an adverse event (AE) of the intervention.

- Time: from date of 1st or 2nd dose of vaccine administered until 12 months thereafter or last follow-up available.

- Study Designs: RCT (phases I, II or III) and cohort studies. Ecological studies will be excluded, and case-control studies may be included in an exploratory analysis.

## Methods

We will follow Preferred Reporting Items for Systematic Review and Meta-Analysis (PRISMA) and have followed the PRISMA-Protocol checklist for the current protocol, see S1 Checklist [25, 26].

## Eligibility criteria

Studies will be selected according to the criteria outlined below.

**Study designs.** We will include RCTs, as well as prospective and retrospective comparative cohort studies. Cohorts with one single arm, and ecological studies will be excluded. Case-control studies will be assessed for pertinent design and validity, and may be included in an additional exploratory analysis.

**Participants.** Studies involving patients aged 12 years and older in any country for whom we have vaccination and outcome information are eligible. We will include studies addressing specific or vulnerable populations. The intervention population will include patients who have received a vaccine of any form.

**Interventions.** The interventions of interest are treatment with any SARS-CoV-2 vaccine approved for use in any country, or completed phase III RCT SARS-CoV-2 vaccine authorized for limited use in any country. A list of vaccines that are currently in either category is provided in Appendix I in S1 Appendix.

**Comparators.** The comparators are either patients receiving a placebo, or participants with no exposure to a SARS-CoV-2 vaccination.

**Outcomes.** There are three outcomes of interest:

1. All-cause mortality (our primary outcome) after the first dose of vaccine, and second dose when reported, whether primary or secondary outcome in original study.

2. COVID-related death (our secondary outcome), whether primary or secondary outcome in original study 14 days after the immunization regimen is complete.

3. Death as an AE after intervention (our secondary outcome), whether primary or secondary outcome in original study from the date of the first dose.

All-cause mortality will encompass all deaths, regardless of cause. COVID-related death will be defined as a participant with a cause of death directly attributable to COVID-19 or a complication thereof as defined by the WHO International Guidelines for Certification and Classification (CODING) of COVID-19 as cause of death. Death as an adverse event will be defined as any death related to the intervention as defined by the WHO COVID-19 Vaccines safety Surveillance Manual [27] and The Global Manual on Surveillance of Adverse Events Following Immunization [28]. Definitions of fatal adverse events will be documented for each study during the data extraction phase. All deaths occurring within one, two, and three months of the first or second dose will be reviewed to determine whether it is considered an AE.

We will include studies with any measures of effect and calculate hazard ratios.

**Timing.** We will include all studies regardless of follow-up duration. Primary outcome will be all-cause mortality up to 12 months following the first vaccination dose, and second vaccination dose when reported, starting from the date of vaccination. For COVID-related death, the timeframe will begin at 2 weeks following the completed regimen and end 12 months later, or until the last follow-up. Death as an AE will be limited to three months following the vaccination, starting from the date of the first dose, or from the second dose if documented. We will extract information by month ($1^{st}$, $2^{nd}$, and $3^{rd}$) after the first dose or, when available, the second dose or booster shot. These timeframes may be modified if found to be incompatible with most articles being reviewed.

**Setting.** There will be no restrictions by type of setting.

**Language.** As excluding languages could ignore key data and potentially introduce bias, we will include relevant articles reported in any language for which we have a translator. Those relevant titles we cannot translate will be provided as an S1 Appendix.

## Search strategy

We will invite a health sciences librarian with experience conducting systematic review searches to aid in the refinement of our strategy. In conjunction with a librarian, we will develop a literature search strategy employing both medical subject headings (MeSH) and text words related to and synonymous with COVID-19, SARS-CoV-2, vaccines, immunotherapy, death, adverse event, treatment outcome, efficacy, effectiveness. The following databases will be searched: Ovid MEDLINE, Embase, PubMed, Web of Science, LILACS and the WHO Global research for COVID-19 database. We will also search for clinical trials and currently in-progress research via: ClinicalTrials.gov, Ovid MEDLINE In-Progress and Other Non-Indexed Citations, EudraCT, Cochrane Library and COVID-19 Study Register, COVID-19 Vaccine Trials (https://covid19vaccinetrial.co.uk/), and medRxiv. The online database search will be supplemented with relevant articles found in the reference lists of eligible articles and pertinent reviews. Grey literature will also be consulted, including searching regulatory and reporting agencies (European Medicines Agency, Pharmaceutical Safety and Efficacy Assessment Report, CDC, WHO), and sponsors and developers will be contacted when deemed appropriate.

Based on a preliminary search of relevant articles conducted by the authors, we developed a systematic search strategy based on various synonyms for the keywords above. We have included the draft search strategy as Appendix II in S1 Appendix. Following completion of the initial search strategy, we will adapt the syntax according to the specific functions of each subsequent database. As research in this field is limited but growing, and we are including observational and controlled studies, our goal is to maximize the sensitivity and broadness of our search. To this end, the search strategy will not include study design filters or population parameters, nor will it exclude any date ranges. In addition, we propose both a provision for repeating searches towards the end of the review process, as well as initiating current awareness alerts.

Once the search strategy is established, one reviewer (SLC) will conduct the search and documentation. The PRISMA-S 16-item checklist for reporting search strategies will be followed. EndNote software will be used for citation management.

## Study selection

Abstracts and titles yielded from the database searches will be uploaded to the free web tool Rayyan to facilitate the screening and selecting of titles. To avoid potential bias stemming from duplicate study inclusion, the reviewers will cross-reference author names as well as juxtapose sample sizes, treatments, and outcomes. Full texts will be handled with the citation software EndNote. Two independent reviewers (SLC and TZ) will conduct the initial screening for inclusion, adhering to predetermined eligibility criteria. If one reviewer deems the article eligible, both reviewers will examine the full text. A PRISMA flow diagram will be created following the screening process. During the full text review phase, the independent reviewers will decide on eligibility. A third independent reviewer (AT) will resolve disagreements. If more data is necessary for a decision, article authors will be contacted. Only if there is consensus on an article's eligibility will data be extracted. Reasons for exclusion will be recorded. We will not blind reviewers to authors and journals.

## Data extraction

We will develop a standardized electronic data extraction form tailored to our review. Prior to extracting data, the form will be piloted on 10 articles by two independent reviewers (SLC and TZ) to ensure the form consistently captures all relevant data. Forms will be calibrated as

necessary. Data extraction will occur in duplicate by the two reviewers. A third reviewer (AT) will arbitrate to resolve disagreements. Article authors will be contacted if there remains uncertainty about data.

Data to be extracted will consist broadly of general study information, study characteristics, patient characteristics, intervention and COVID-related data, outcome/results, and additional information not previously classified. We will also take note of the study period to check for the epidemiological situation in the country (background risk of disease and death, variants of concern). If not reported in the original study this will be verified in publicly available statistics. A precursory listing of data we intend to extract is included as Appendix III in S1 Appendix. While this list is based on data we expect to find based on available checklists and preliminary articles reviewed, it is not exhaustive. We will modify the form where necessary after the search is completed, and prior to beginning data extraction. One data extraction form will be used for all articles included.

## Assessment of methodological quality

Two reviewers (AT and TZ) will independently perform the quality assessment, with any discrepancy being resolved by a third reviewer (GLW). The agreement between reviewers will be expressed as a percentage. The assessment will be blinded for the third reviewer. The reviewers will use two separate tools to assess risk of bias, depending on study design. For the RCTs, we will use the Cochrane Risk of Bias 2 tool [29]. We will assess risk of bias according to the following domains: random sequence generation, allocation concealment, blinding (subjects, personnel, outcome assessment), incomplete outcome data, selective outcome reporting, and source of funding [30].

Quality of observational studies will be assessed using the ROBINS-1 tool [30]. We will evaluate risk of bias due to confounding, selection, interventions (classification/deviations), missing data, measurement of outcomes and selection of the reported result. The confounders will be determined based on the WHO's interim guidance on assessing COVID-19 vaccine effectiveness using observational study designs [19]. As such, we are expecting to consider exposure and background risk (including variants of concern circulating in each setting), prior SARS-CoV-2 infection, health care seeking behavior, risk reduction behavior, healthcare access, age, sex, comorbidities, and time since vaccination amongst others. An emphasis will be placed on age and background risk—including variants of concern—in our discussion.

We will judge each potential source of bias as high, low, or unclear and provide justification for our judgement.

Sensitivity analyses will be done excluding low-quality studies in scenario 1. In a second scenario, we will explore how low-quality studies would have affected the results.

The certainty of evidence will be rated using the GRADE approach (considering risk of bias, consistency of effect, imprecision, indirectness, and publication bias) [30].

## Data analysis

We will pool RCT and observational studies separately in our primary analysis and, if possible, altogether in secondary analyses. Case-control studies will be examined in an exploratory analysis. We will analyze data as hazard ratio (HR) providing 95% confidence intervals (CI) for the results under a proportional hazards model for the exposure. The effect of vaccination on all-cause mortality as well as secondary outcomes will be expressed as (1-HR). HR will be calculated, when possible, using the methods described by Tierney [31]. If multiple studies are found reporting risk ratios (RR) and odds ratios (OR) instead of HR, RR and OR will be converted to HR using the methods described by VanderWeele [32].

We will not adjust HR in RCT. Adjusted values will be used in observational studies if appropriate and if the same confounders and methods of adjustment are found across studies. Additionally, we will consider a sensitivity analysis using adjusted values based on the study-specific adjustments. We will preferably use age adjustments. If there are many null values, the approach outlined by Negeri et al. will be applied [33]. We will assess clinical, methodological, and statistical heterogeneity. Assessment of statistical heterogeneity amongst the studies will be done using $I^2$ and prediction intervals, as well as visual inspection of the forest plots. The following thresholds will be used for the $I^2$ test:

0% to <30%: might not be important;

30% to <50%: may represent moderate heterogeneity;

50% to <75%: may represent substantial heterogeneity;

75% to 100%: considerable heterogeneity [25].

We will investigate the sources of heterogeneity if the threshold of 75% has been crossed. If considerable heterogeneity is found, we will use a narrative approach towards data synthesis. If studies are found to be homogeneous, a meta-analysis for each study design (RCT/cohort) will be done using the Meta package of the R studio software. When feasible, we will perform pre-specified subgroup analyses according to type of vaccine (mRNA, viral vector, inactivated, protein subunit) and study country. In addition, we will perform a meta-regression if enough power and sufficient data are available for the following variables of potential interest: age, sex, time since latest dose, number of doses, vaccine type, variants circulating and availability to of viral gene sequencing in the country, previous SARS-CoV-2 infection, comorbidities, exposure and background risk, health care seeking behavior, risk reduction behavior, healthcare access, and setting (community, long-term care, hospital). If we are unable to do so we will prioritize meta-regression for age, number of doses, time since latest dose, variants of concern, and background risk. A similar approach will be undertaken in our exploratory analysis of case-control studies.

As we expect to find clinical and statistical heterogeneity, we will use a random-effects model using the Der Simonian & Laird method in our primary analysis. A fixed-effect model (Mantel-Haenszel method) will be used as a sensitivity analysis.

Publication bias will be assessed using a funnel plot and Egger statistical test if we have more than 10 studies in a forest plot.

Finally, we plan to conduct the review according to this protocol but will report any deviations. Reporting of the review will follow PRISMA-2020 guidelines [34].

The protocol is registered at Prospero (York University), # CRD42021262211.

## Supporting information

**S1 Checklist. PRISMA-P 2015 checklist.**
(PDF)

**S1 Appendix. Appendices I, II, III.**
(DOCX)

## Acknowledgments

We would like to thank Dr. Dick Menzies, M.D. RECRU, Departments of Medicine and of Epidemiology, Biostatistics & Occupational Health, McGill University, Montreal and Nicholas Winters, M. Sc., Ph.D candidate Departments of Medicine and of Epidemiology, Biostatistics & Occupational Health, McGill University, Montreal for their assistance and support in the elaboration of this protocol.

## Author Contributions

**Conceptualization:** Anete Trajman, Sophie Lachapelle-Chisholm, Théodora Zikos.

**Data curation:** Sophie Lachapelle-Chisholm.

**Formal analysis:** Théodora Zikos, Andrea Benedetti.

**Investigation:** Sophie Lachapelle-Chisholm.

**Methodology:** Anete Trajman, Théodora Zikos, Guilherme Loureiro Werneck, Andrea Benedetti.

**Writing – original draft:** Anete Trajman, Sophie Lachapelle-Chisholm, Théodora Zikos.

**Writing – review & editing:** Anete Trajman, Sophie Lachapelle-Chisholm, Théodora Zikos, Guilherme Loureiro Werneck, Andrea Benedetti.

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
