## [Decision Letter · Decision Letter 0]

21 Oct 2021

PONE-D-21-21550Efficacy and effectiveness of SARS-CoV-2 vaccines for death prevention: A protocol for a systematic review and meta-analysis

PLOS ONE

Dear Anete Trajman,

Thank you for submitting your manuscript to PLOS ONE. After careful consideration, we feel that it has merit but does not fully meet PLOS ONE’s publication criteria as it currently stands. Therefore, we invite you to submit a revised version of the manuscript that addresses the points raised during the review process.

Please consider the comments by reviewer #2.

As the comments are minor, please submit your revised manuscript by 30 October 2021. If you will need more time than this to complete your revisions, please reply to this message or contact the journal office at plosone@plos.org. Please include the following items when submitting your revised manuscript:

We look forward to receiving your revised manuscript.

Kind regards,

Hemant Deepak Shewade, MBBS MD PhD

Academic Editor

PLOS ONE

2. We note that this manuscript is a systematic review or meta-analysis; our author guidelines therefore require that you use PRISMA guidance to help improve reporting quality of this type of study. Please upload copies of the completed PRISMA checklist as Supporting Information with a file name “PRISMA checklist”.

Reviewers' comments:

Reviewer's Responses to Questions

**Comments to the Author**

1. Does the manuscript provide a valid rationale for the proposed study, with clearly identified and justified research questions?

Reviewer #1: Yes

Reviewer #2: Yes

2. Is the protocol technically sound and planned in a manner that will lead to a meaningful outcome and allow testing the stated hypotheses?

Reviewer #1: Yes

Reviewer #2: Yes

3. Is the methodology feasible and described in sufficient detail to allow the work to be replicable?

Reviewer #1: Yes

Reviewer #2: Yes

4. Have the authors described where all data underlying the findings will be made available when the study is complete?

Reviewer #1: Yes

Reviewer #2: Yes

5. Is the manuscript presented in an intelligible fashion and written in standard English?

Reviewer #1: Yes

Reviewer #2: Yes

6. Review Comments to the Author

You may also provide optional suggestions and comments to authors that they might find helpful in planning their study.

Reviewer #1: A well structured protocol which bring out the gap in analysis of the existing vaccine efficacy and effectiveness. The proposed protocol may address the gap in analyzing the all cause mortality related to covid-19 vaccination as well as the protection offered by Covid-19 vaccination in preventing deaths.

Reviewer #2: The authors have written the protocol to study a very critical estimate with respect to COVID-19 vaccines. I thank the authors for this study as its essential in coming months and years to come. I am really excited and look forward to the results from this review. Wish them all the best. I have few suggestions and comments:

1) The authors can also add the WHO Global research for COVID-19 database in the search strategy. It’s exhaustive and covers all languages across the world.

2) For reference 6, I would also recommend looking into the LHSTM vaccine landscape as it started well before WHO initiated its dashboard. This reference would also be useful to pool results of different vaccine trials across different studies.

3) Apart from citing reference 18, I would also consider non availability of timely genomic sequencing data across several countries to be a confounder as vaccines tend to show different efficacy/real world effectiveness across strains. I would be interested to see whether the authors would consider a sub-group analysis on this aspect. (Agree that this data is not well published across all countries and can be a potential limitation in this study as well as core recommendation for global public health policy).

4) Reference 22 is a potential cross validation to the results you will be getting in your research.

5) By mentioning cohort studies, I believe vaccine challenge studies would also be included.

6) By mentioning case-control studies, I believe test negative studies would be included.

7) The Disease definition needs to be looked into very carefully as it determines the end point of the review. Several countries tend to use different cut off points for cT values. It would be interesting to see this in sub-group analysis (if the cut-off is mentioned in the study).

8) After reading the objectives, I get a sense that only research articles would be reviewed. There are countries like India that show deaths between vaccinated and non-vaccinated in a public website. Will such sources be included?. I stress this because as the authors mentioned, most studies see primary outcomes like hospitalization, Oxygen use, ICU admission, etc.

9) I think for all-cause mortality a cut-off for 4 weeks would be ideal as this is how post-illness sequelae deaths are classified in Influenza and other respiratory viruses. This cut –off was played around by several countries to get their reported death numbers down.

10) I noticed you had mentioned that all-cause mortality was taken primarily due to the poor COVID-19 attribution/certification in several countries. How do you plan to address the poor classification of death as an adverse event in several countries?

11) Suggest using Zotero instead of EndNote. Chances of duplication are there in EndNote. This was my experience. I leave the choice to the authors.

12) I would like to have more details on the electronic data extraction form. What are the added benefits of using this as compared to traditional methods ?

13) Point 3 is addressed in your methods. Thank you

14) I am surprised why low quality studies are excluded and then included in sensitivity analysis. I get the authors thought behind this. You can rephrase the statement as “sensitivity analysis would be done among the included studies in scenario 1. In scenario 2 we would explore how low quality studies would have affected the results.

7. PLOS authors have the option to publish the peer review history of their article (what does this mean?). If published, this will include your full peer review and any attached files.

Reviewer #1: No

Reviewer #2: **Yes: **Giridara Gopal Parameswaran

---

## [Author Response · Author response to Decision Letter 0]

6 Jan 2022

To the editor 

Editor Plos One 

Dear editor and reviewers 

Thank you for the opportunity to improve our manuscript based on the useful suggestions and insights from the reviewers. You will find below a point-by-point reply to each of their comments. We have also updated a couple of information on the Introduction session and on the Methods sessions (and appendix III), as knowledge and practice (boosters, for example) is advancing quickly. We hope the manuscript is now suitable for publication. 

Yours truly 

Anete Trajman, on behalf of the authors. 

Reviewer 2: 

The authors have written the protocol to study a very critical estimate with respect to COVID-19 vaccines. I thank the authors for this study as its essential in coming months and years to come. I am really excited and look forward to the results from this review. Wish them all the best. I have few suggestions and comments: 

R: Thank you for your encouragement, and for your suggestions. 

1) The authors can also add the WHO Global research for COVID-19 database in the search strategy. It’s exhaustive and covers all languages across the world. 

R: That is an excellent suggestion, we will search this comprehensive database. 

2) For reference 6, I would also recommend looking into the LHSTM vaccine landscape as it started well before WHO initiated its dashboard. This reference would also be useful to pool results of different vaccine trials across different studies. 

R: We have consulted the LHSTM Covid-19 vaccine landscape and added as a reference. 

3) Apart from citing reference 18, I would also consider non availability of timely genomic sequencing data across several countries to be a confounder as vaccines tend to show different efficacy/real world effectiveness across strains. I would be interested to see whether the authors would consider a sub-group analysis on this aspect. (Agree that this data is not well published across all countries and can be a potential limitation in this study as well as core recommendation for global public health policy). 

R: We agree VoC are a major confounder – and your comment could not be timelier, given the emergence of the Omicron variant probably in Botswana but its discovery in South Africa, where there are high-level researchers and labs sequencing the virus. We have added to the text but honestly, we doubt we will be able analyze by this (and other) subgroups. It is certainly worth a try, and we will extract the data. 

4) Reference 22 is a potential cross validation to the results you will be getting in your research. 

R: They have an ambitious project! The number of individual patients is huge now, and I wish them the best of luck. We are eager to hear from them. 

5) By mentioning cohort studies, I believe vaccine challenge studies would also be included. 

R: Vaccine challenge studies usually use antibody of other marker response as the main outcome. We will include them to check if there were any reported deaths and try to extract the meaningful data. 

6) By mentioning case-control studies, I believe test negative studies would be included. 

R: Study selection will be based on their methods, any results of included studies will be considered. 

7) The Disease definition needs to be looked into very carefully as it determines the end point of the review. Several countries tend to use different cut off points for cT values. It would be interesting to see this in sub-group analysis (if the cut-off is mentioned in the study). 

R: That is an interesting point. We have added this variable in our list of tentative data extraction (Appendix III). If not available in the studies, we will try to find the information with authors or country MoH. 

8) After reading the objectives, I get a sense that only research articles would be reviewed. There are countries like India that show deaths between vaccinated and non-vaccinated in a public website. Will such sources be included?. I stress this because as the authors mentioned, most studies see primary outcomes like hospitalization, Oxygen use, ICU admission, etc. 

R: We can see below that later in the Methods session it became clear that any publicly available routine database will be included, as long as the analyses meet the criteria (e.g., ecological country analyses will not be included). With regards to the grey literature, we did not include this data. Many countries have published (in highly prestigious scientific journals) their routine data, as the approach was sufficiently rigorous, and indeed some cohort studies have been published by some countries, e.g.: 

- Haas, E.J., Angulo, F.J., Mclaughlin, J.M., Anis, E., Singer, S.R., Khan, F., Brooks, N., Smaja, M., Mircus, G., Pan, K., Southern, J., Swerdlow, D.L., Jodar, L., Levy, Y., Alroy-Preis, S., 2021. Impact and effectiveness of mRNA BNT162b2 vaccine against SARS-CoV-2 infections and COVID-19 cases, hospitalisations, and deaths following a nationwide vaccination campaign in Israel: an observational study using national surveillance data. The Lancet doi:10.1016/s0140-6736(21)00947-8 

-Madhumathi Ramakrishnan, Prakash Subbarayan, Impact of vaccination in reducing Hospital expenses, Mortality and Average length of stay among COVID-19 patients – a retrospective cohort study from India 

-medRxiv 2021.06.18.21258798; doi: https://doi.org/10.1101/2021.06.18.21258798

-Jara A, Undurraga EA, González C, Paredes F, Fontecilla T, Jara G, Pizarro A, Acevedo J, Leo K, Leon F, Sans C, Leighton P, Suárez P, García-Escorza H, Araos R. Effectiveness of an Inactivated SARS-CoV-2 Vaccine in Chile. N Engl J Med. 2021 Jul 7. doi: 10.1056/NEJMoa2107715. Epub ahead of print. PMID: 34233097. 

- Macchia A, Ferrante D, Angeleri P, et al. Evaluation of a COVID-19 Vaccine Campaign and SARS-CoV-2 Infection and Mortality Among Adults Aged 60 Years And Older in a Middle-Income Country. JAMA Netw Open. 2021;4(10):e2130800. doi:10.1001/jamanetworkopen.2021.30800 

9) I think for all-cause mortality a cut-off for 4 weeks would be ideal as this is how post-illness sequelae deaths are classified in Influenza and other respiratory viruses. This cut –off was played around by several countries to get their reported death numbers down. 

R: That is an interesting suggestion. As noted in our manuscript, we will investigate all-cause mortality up to 12 months, but will extract and analyze the information by month (1st, 2nd, 3rd) after the first or second dose, or booster shot. 

10) I noticed you had mentioned that all-cause mortality was taken primarily due to the poor COVID-19 attribution/certification in several countries. How do you plan to address the poor classification of death as an adverse event in several countries? 

R: We will not judge the classification; we will deal with its eventual poor methods by using all-cause mortality. Indeed, it would be very complex to distinguish quality of this classification. For example, Peru deliberately used excess deaths as the measure of Covid death, as they were unable to establish a proper diagnosis in the peak of the pandemics. This is exactly why we chose to use any death as the outcome. We felt that this was the way we would best appreciate the outcome, although we agree that there are limitations in this choice (other alternatives would have other limitations). 

11) Suggest using Zotero instead of EndNote. Chances of duplication are there in EndNote. This was my experience. I leave the choice to the authors. 

R: We agree, we do plan to use EndNote for the SR (see last sentence of the Search Strategy session). 

12) I would like to have more details on the electronic data extraction form. What are the added benefits of using this as compared to traditional methods? 

R: By electronic data extraction form we mean an online form, such as a shared excel file. This will facilitate comparison of results between the data extractors, and ensure it is safely filed and properly documented. 

13) Point 3 is addressed in your methods. Thank you 

14) I am surprised why low quality studies are excluded and then included in sensitivity analysis. I get the authors thought behind this. You can rephrase the statement as “sensitivity analysis would be done among the included studies in scenario 1. In scenario 2 we would explore how low quality studies would have affected the results. 

R: Thank you for your suggestion. We accepted and have rephrased.

---

## [Decision Letter · Decision Letter 1]

2 Mar 2022

Efficacy and effectiveness of SARS-CoV-2 vaccines for death prevention: A protocol for a systematic review and meta-analysis

PONE-D-21-21550R1

Dear Sophie Lachapelle-Chisholm,

We’re pleased to inform you that your manuscript has been judged scientifically suitable for publication and will be formally accepted for publication once it meets all outstanding technical requirements.

Kind regards,

Hemant Deepak Shewade, MBBS MD PhD

Academic Editor

PLOS ONE

Additional Editor Comments (optional):

Reviewers' comments:

Reviewer's Responses to Questions

**Comments to the Author**

1. Does the manuscript provide a valid rationale for the proposed study, with clearly identified and justified research questions?

Reviewer #2: Yes

2. Is the protocol technically sound and planned in a manner that will lead to a meaningful outcome and allow testing the stated hypotheses?

Reviewer #2: Yes

3. Is the methodology feasible and described in sufficient detail to allow the work to be replicable?

Reviewer #2: Yes

4. Have the authors described where all data underlying the findings will be made available when the study is complete?

Reviewer #2: Yes

5. Is the manuscript presented in an intelligible fashion and written in standard English?

Reviewer #2: Yes

6. Review Comments to the Author

You may also provide optional suggestions and comments to authors that they might find helpful in planning their study.

Reviewer #2: Look forward for this interesting study and how the results pan out. I will be seeing on the broader thought whether the hypothesis of irregularly timed/phased vaccination strategies actually led to rise new variants and poor vaccine efficacy.

7. PLOS authors have the option to publish the peer review history of their article (what does this mean?). If published, this will include your full peer review and any attached files.

Reviewer #2: **Yes: **Giridara Gopal Parameswaran

---

## [Editor Report · Acceptance letter]

15 Apr 2022

PONE-D-21-21550R1 

Efficacy and effectiveness of SARS-CoV-2 vaccines for death prevention: A protocol for a systematic review and meta-analysis 

Dear Dr. Lachapelle-Chisholm:

I'm pleased to inform you that your manuscript has been deemed suitable for publication in PLOS ONE. Congratulations! Your manuscript is now with our production department. 

Kind regards, 

on behalf of

Dr. Hemant Deepak Shewade 

Academic Editor

PLOS ONE